# Simulation of Drilling Temperature Rise in Frozen Soil of Lunar Polar Region Based on Discrete Element Theory

**Jinsheng Cui** [1], **Le Kui** [1], **Weiwei Zhang** [2,\*], **Deming Zhao** [3] **and Jiaqing Chang** [1]

1   School of Mechanical and Electric Engineering, Guangzhou University, Guangzhou 510006, China
2   School of Mechatronics Engineering, Harbin Institute of Technology, Harbin 150001, China
3   School of Mechanical Engineering, Zhejiang Sci-Tech University, Hangzhou 310018, China
\*   Correspondence: zweier@hit.edu.cn

**Abstract:** As the frozen soil in the South Pole region of the Moon is an important water resource, the operation of drilling and retrieving samples of the frozen soil in this region will be a crucial task for us to accomplish in future deep-space exploration. Thus, this paper investigated the effects of the increasing temperature and heat transfer between the drilling tools and the simulated lunar soil to minimize the degradation of the frozen soil samples during drilling due to the increased temperature. Specifically, the discrete element method was adopted and the heat transfer parameters of the discrete element particles were calibrated based on the equivalent heat transfer of the particle system. Moreover, a lunar soil particle system was developed for the simulations. Under the current working conditions with reasonable parameters, the maximum increase in the drill bit temperature was about 60 °C. Overall, the simulation results were consistent with the experimental results, and further analysis revealed that the flow of lunar soil can effectively take away thermal, which is also one of the reasons why the simulated lunar soil particles are in a high-temperature state at the front of the drilling tool.

**Keywords:** planetary drilling; lunar polar frozen soil; temperature rise; discrete element method

## 1. Introduction

The Chang'e-7 lunar mission plans to employ a drilling method to obtain in situ frozen soil samples and analyze their composition. The drilling of lunar soil using the cutting edge of a rotary drill bit is a complicated mechanical tool–soil coupling process. However, due to the high vacuum and the low thermal conductivity of the moon, acquiring frozen soil samples from its polar region via drilling generates extensive thermal that causes the samples to lose the original characteristic information. Thus, extensive research has been conducted on this topic.

Liu et al. [1] proposed a corresponding scheme to design the task operation mode and sampling machine for the drilling of frozen soil in the polar region based on the analysis of the sampling environment and the object in the polar region. Furthermore, Liu et al. [2] prepared and drilled simulated lunar soil with varying water content. Wang et al. [3] investigated the distribution characteristics and existing forms of lunar frozen soil resources and proposed an integrated development and utilization scheme for photothermal drilling. Wood [4] proposed a model to evaluate the effective thermal conductivity of the planetary regolith or a system of porous particles composed of solid particles and gases. Chen et al. [5] used the discrete element method (DEM) to determine the motion and stress characteristics of the particles transported by an auger during work. Zhang et al. [6] revealed a complex and universal mechanism to simulate heat transfer in the particulate gases in lunar soil, which can be used to construct an experimental environment for conducting scientific research on the lunar soil.

In recent years, DEM has been widely used to research of the heat transfer characteristics of materials. DEM provides an effective way to study the drilling thermal

characteristics of lunar soil through numerical simulation. Li et al. [7] introduced a Polygon Contact Model into the DEM method and applied four different contact force models into the newly proposed DEM algorithm to analyze their differences and implications. Vargas and McCarthy [8–11] introduced the DEM contact heat transfer model, developed the dynamic hot-particle method, and analyzed the heat transfer mechanism in particles. In addition, Cui et al. [12] employed the DEM to simulate the heat generation in lunar rock drilling. Liu et al. [13] investigated the influence of particles of varying sizes on the axial force and torque of the drill tool to prevent blockage at the front of the drill tool during drilling. Gong et al. [14] designed a scheme to predict the effective thermal conductivity of granular materials composed of a homogeneous matrix. This scheme explores the influences that particle shape and packing density have on the effective thermal conductivity of materials. Liang et al. [15] studied heat conduction between particles and the convective heat transfer between gas and particles during the heat transfer of stacked particles. Wu et al. [16] proposed a method to predict the correlation between the factors of radiation exchange under various porosities, focusing on the relationship between the effective thermal conductivity of radiation, temperature, and porosity. Tang et al. [17] established a two-dimensional heat transfer model for a two-vacancy particle bed in order to determine the effects of particle size and thermal conductivity on the heat transfer of the accumulated particles. Lee et al. [18] and Calvet et al. [19] proposed a method that combined the discrete and finite elements to calculate the effective thermal conductivity of the medium between particles. Govender et al. [20] examined the influence of the particle shape on the effective thermal conductivity and heat distribution of a particle system. Xiao [21] proposed an empirical equation relating to the particle shape to quantify the influence of relative density and particle distribution on the thermal conductivity of particles in the experiment. Chen et al. [22] studied the thermal DEM model of various materials with different particle sizes and derived a thermal contact theory to calculate the comprehensive heat transfer. Akhil et al. [23] assessed the influence of a particle system with a single-size under uniaxial pressure on the effective thermal conductivity at various clearance and strain cutoff ranges.

## 2. Model

### 2.1. Heat Transfer Model

2.1.1. Heat Transfer Model between Particles

In the drilling process, the frozen lunar soil is composed of simulated lunar soil particles, water ice, and an air medium, wherein the simulated lunar soil particles are the primary constituents. Therefore, in the simulation of the discrete element software, the frozen lunar soil is considered a particle system. The heat transfer modes of the lunar frozen soil include heat conduction, convection heat transfer, and radiation heat dissipation. For a particle system, the discrete element involves no convection or radiation model. In this study, these three heat transfer modes are equivalent to the effective heat conduction, represented by the effective heat conduction coefficient that is used in calculating the heat transfer of the particle system. Specifically, the method of evaluating heat conduction between two elements in contact with a discrete element can be stated as [10]

$$Q_{ij} = 2k_s \left( \frac{3F_n r^*}{4E^*} \right)^{1/3} (T_j - T_i) \tag{1}$$

where $Q_{ij}$ denotes the energy transferred from particle j to particle i, $k_s$ denotes the thermal conductivity of the granular material, $F_n$ indicates the normal contact force between particles, and $E^*$ denotes the equivalent Young's modulus between the particles, where $E_i$ and $E_j$ represent the Young's modulus of particles i and j, respectively; $v_i$ and $v_j$ denote the Poisson's ratios of particles i and j, respectively. $r^*$ indicates the equivalent radius of the particle, where $r_i$ and $r_j$ denote the radii of particles i and j, respectively; $T_i$ and $T_j$ denote the temperatures of particles i and j.

Regarding the effects of simulated lunar soil radiation, the heat transfer between particles calculated according to Equation (1) only mainly represents the heat conduction between particles, and it seems that convection and radiation are not considered at all. On the one hand, the main reason is that the heat transfer between particles plays a leading role. On the other hand, in order to simplify the model, all heat transfer processes between particles are generally simplified as heat conduction in DEM, which is called effective heat conduction and characterized by effective thermal conductivity (ETC). In the subsequent calibration process of particle parameters, the goal is to make the simulated particle set similar to the measured effective thermal conductivity of the actual simulated lunar soil. In fact, for a particle system, the experimental thermal conductivity is the ETC under the experimental conditions. To sum up, considering the simplification and error of the model, the present study focused on the ETC in the investigation of heat transfer through a granular assembly.

### 2.1.2. Heat Transfer Model of Geometry

In addition to the particle, the entity in contact with the particle in the discrete element is referred to as the geometry, e.g., a drilling tool. In this study, the heat transfer model for calculating the temperature of the drill tool according to Cui et al. [24]. We assumed that the heat source was located at the front of the drill tool during the drilling process. The drill tool was semi-infinite in the axial direction, and the temperature of each cross-section remained constant. Considering the entire drilling process in terms of energy input, the calculation steps are detailed as follows:

1. The drill tool is discretized into equispaced drill tool elements and segmented into time step;
2. In the current time interval, the variations of the drill temperature caused by the heat source at the front of the drill were calculated, and the influence of convective heat transfer and radiation heat dissipation on the heat transfer inside the drill is considered;
3. In the current time interval, the original continuous temperature field was no longer continuous because of radiation or convection, and thermal was transferred from the high-temperature element to the low-temperature element inside the drill tool. Herein, this process is referred to as the secondary heat conduction. Moreover, secondary heat conduction was calculated using the central difference method;
4. In the subsequent time interval, steps 2 and 3 are repeated;
5. End of simulation and output results.

### 2.2. *Discrete Element Model*

### 2.2.1. Particle Modeling

If the particle size range of the actual lunar soil is used to construct a particle system model, the number of simulated lunar soil particles within the system will be huge. In the discrete element, the number of particles is an essential factor affecting the simulation speed. Therefore, the diameter of the lunar soil particle adopted in the developed particle system for the simulations was larger than the actual value. To ensure the effectiveness of the subsequent simulation, the particle diameter should not be excessively large. Based on the above requirements, this study adopted the variable particle diameter modeling method [25] to establish the lunar soil particle system for simulations. The variable particle diameter modeling refers to the interior segmentation of the particle system into several areas, which increases the diameter of the particles from the inside to the outside to reduce the number of particles and improve the calculation speed. In this study, the particle system was classified into three areas. The first area was in close contact with and around the drill tool. The diameters of the particles in the outside area were two and three times the diameter of the particles inside. The three-stage particle system with variable diameters is illustrated in Figure 1.

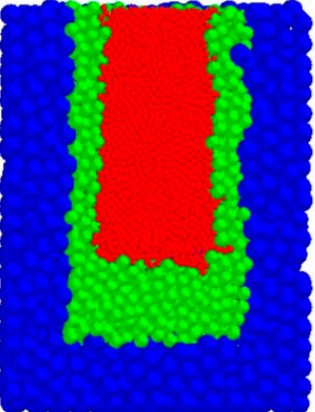

**Figure 1.** Three-stage variable particle diameter within the particle system.(The red zone has a particle radius of 0.8 mm, the green zone has a particle radius of 1.6mm, and the blue zone has a particle radius of 2.4 mm).

In the discrete element, the contact model between the particles forms the core component of the discrete element algorithm, which determines the behavior between the contact particles after contact (contact model in the discrete element neglects the contact between geometry), including the variations in dynamic, temperature, charge, etc. The Hertz–Mindlin model was employed as the contact model in this study. In the discrete element, this basic contact model calculates the basic forces of the particles in contact. Through the API interface of the discrete element software EDEM, the heat transfer calculation model (Section 2.1.2) was implemented by C++ programming and a dynamic link library file (.dll file) was generated, which was loaded into the discrete element software EDEM to calculate the temperature of the drill tools and particles during drilling.

Based on the above method, a particle system was established in the discrete element software, with a thermal conductivity similar to that of the lunar soil simulated in the experiment through parameter calibration. The center-composite design method employed by Deng et al. [26] was adopted to calibrate the parameters of the particle system, including the particle diameter, thermal conductivity, and Young's modulus. The settings of the particle system parameters after calibration are listed in Table 1. According to ESA survey data, the water content of frozen soil in the lunar polar region is less than 11.9%. According to our preliminary research and analysis results, the water content of lunar frozen soil in the permanently shadowed area, formed by meteorite impact craters at the lunar South Pole, is between 5 and 15% [2]. Therefore, after comprehensive consideration, simulated lunar soils with a water content of 5% and 10%, respectively, were studied in our experiment.

**Table 1.** Parameters of the particle system.

| Water Content (wt) | Effective Thermal Conductivity (W/m·K) | Specific Heat Capacity (J/kg·°C) | Shear Modulus (MPa) |
|---|---|---|---|
| 5% | 50.00 | 228.95 | 40.00 |
| 10% | 66.18 | 406.70 | 40.00 |

### 2.2.2. Geometric Modeling

As drilling tools cannot be directly modeled using EDEM, we developed the drilling tool model using Solidworks and saved it in a specific format into EDEM. The discrete element model used in the simulation is presented in Figure 2. The simulated lunar soil particles were located in lunar soil barrels (disregarded in the heat transfer calculations). Specifically, the established particle system was set to a diameter of 70 mm, and the length of the drill tool was set to 150 mm, with its elements spaced in intervals of 5 mm. The discrete drill tool is depicted in Figure 3.

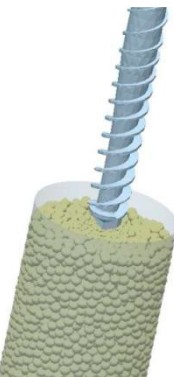

**Figure 2.** Discrete element simulation model drilling simulated lunar soil.

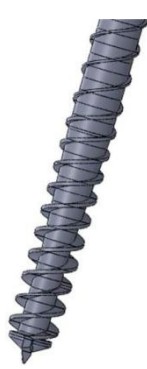

**Figure 3.** Drill tool model after discretization.

## 3. Experimental Apparatus and Simulated Frozen Lunar Soil

### 3.1. Experimental Apparatus

As depicted in Figure 4, the drilling tests were performed in a simulated environment for HIT profile sampling on the drilling efficiency test rig. This apparatus can create a minimum sampling environment of –196 °C and ensure the atmospheric environment in the drilling area through nitrogen to completely isolate the impact of the environment. The simulated samples were obtained using ultra-low-temperature freezers and folic acid refrigeration, and a low-temperature environment was created using nitrogen. In the test process, a Pt100 platinum resistance and K-type thermocouple was used to acquire the real-time data of the drilling tool temperature.

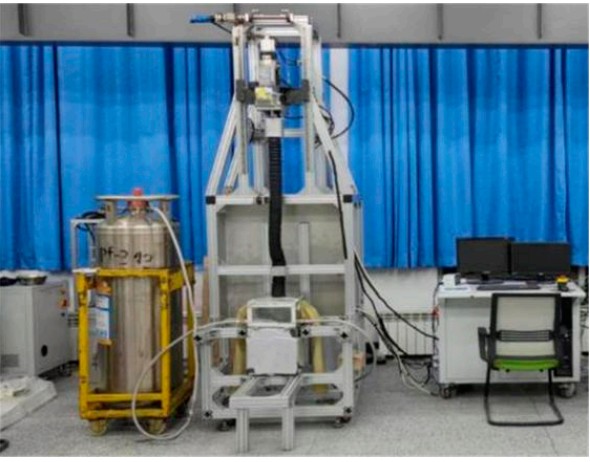

**Figure 4.** Experimental apparatus.

The selection of the temperature measurement point of the drill tool is indicated in Figure 5. In particular, the temperature sensor was placed at 4 mm from the drill bit to monitor and acquire the temperature of the drill tool during drilling.

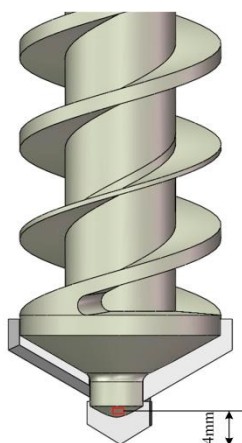

**Figure 5.** Selection of temperature measuring point of drill tool.

*3.2. Simulated Frozen Lunar Soil*

The simulated lunar soil samples are composed of anorthosite and basalt with different particle sizes, as shown in Table 2.

**Table 2.** The minerals, particle size range, and content of simulated lunar soil samples.

| Mineral Class @Proportion of | Particle Size Range | Proportion of |
|---|---|---|
| Anorthosite(A) @(70%) | 0.025–0.05 mm | 31.568% |
| | 0.05–0.075 mm | 6.797% |
| | 0.075–0.1 mm | 10.545% |
| | 0.25–0.5 mm | 10.545% |
| | 0.5–1 mm | 10.545% |
| Basalt(B) @(30%) | 0.025–0.05 mm | 13.502% |
| | 0.05–0.075 mm | 2.920% |
| | 0.075–0.1 mm | 4.526% |
| | 0.25–0.5 mm | 4.526% |
| | 0.5–1 mm | 4.526% |

The simulated lunar soil frozen soil samples were configured according to the following process:

1. Weigh all kinds of particle size anorthosite and basalt, into the oven for drying (more than 8 h);
2. According to the different material different particle size ratio configuration, place into a blender for uniform mixing;
3. After the mixing of dry soil, the mixing of water samples should be allocated according to dry soil and different water content.
4. After the completion of mixed water configuration, homogenize seal stand for 6 to 8 h;
5. Use a press to compact the sample five times to the required compactness;
6. Sample the samples after compaction to verify the actual moisture content of the samples after preparation;
7. Transfer the sample to the secondary refrigeration freezer (−80 °C) for storage after 6–8 h of primary refrigeration (−30 °C), and the sample needs to undergo secondary refrigeration for 6–8 h before use.

Table 3 shows the effective thermal conductivity, specific heat capacity, and compactness of the simulated lunar soil frozen soil particle system measured in the experiment.

**Table 3.** Experimental parameters of simulated lunar soil frozen soil.

| Type | | Parameter | |
|---|---|---|---|
| Particle size range | | 0–1 mm | |
| Sample temperature | | 93 K | |
| Basic mineral | | Pure dry soil sample and mixed water sample | |
| Moisture content | | 5wt% | 10wt% |
| Density (g/cm$^3$) | | 1.9 | 1.75 |
| Measurement result | Effective thermal conductivity(W/(m·K)) | 0.8611 | 1.1397 |
| | Specific heat capacity (J/(kg·°C)) | 228.95 | 270.6 |

## 4. Results and Discussion

### 4.1. Drilling Parameters and Experimental Results

The drilling test was performed on simulated lunar frozen soil with water contents of 5% and 10%, respectively. The parameters for drilling simulated lunar soil with varying water contents are presented in Table 4.

**Table 4.** Drilling parameters.

| No. | Water Content (wt%) | Rotational Speed (rpm) | Feed Rate (mm/min) | Drilling Duration (s) |
|---|---|---|---|---|
| A1 | 5% | 120 | 0.62 | 120 |
| A2 | 5% | 250 | 12.66 | 220 |
| A3 | 10% | 250 | 5.70 | 190 |

In the three groups of drilling experiments, the torque and temperature variation curves of drill tools Nos. A1, A2, and A3 are plotted in Figures 6–8, respectively.

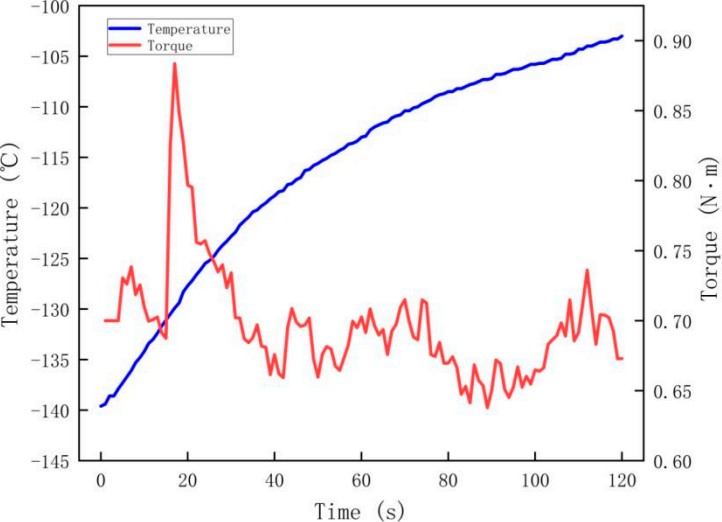

**Figure 6.** Torque and temperature curves of No. A1.

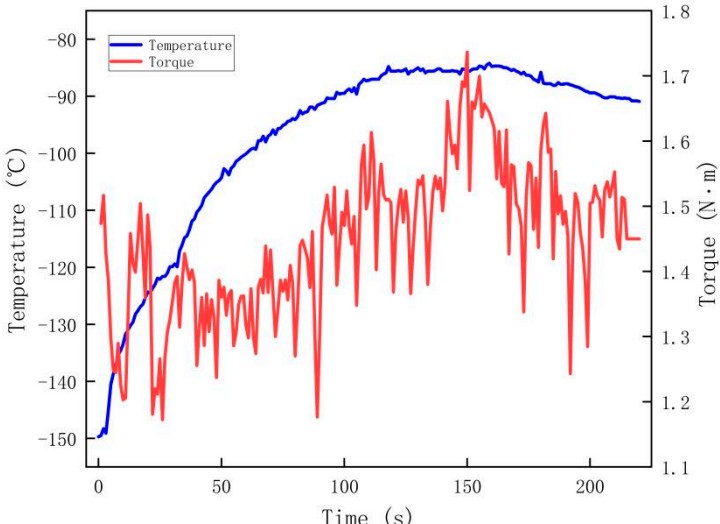

**Figure 7.** Torque and temperature curves of No. A2.

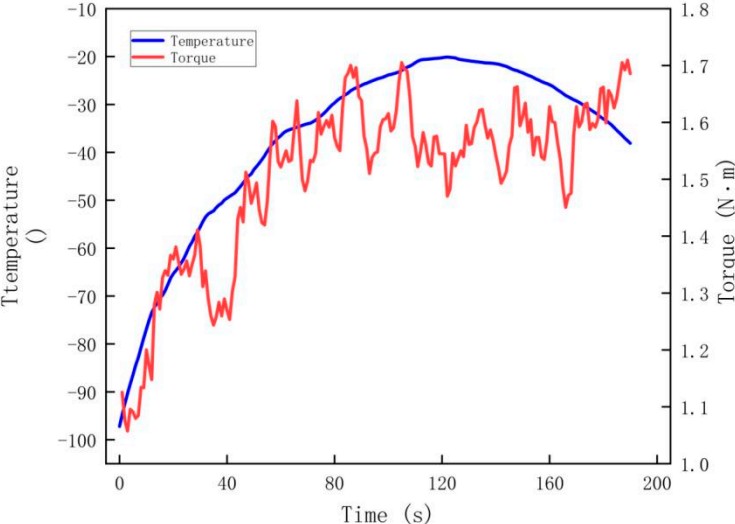

**Figure 8.** Torque and temperature curves of No. A3.

During the experiment, the increase in temperature at the front of the drill tool in Nos. A1, A2, and A3 was 36.60, 58.80, and 59.11 °C, respectively. During drilling, the torque on No. A1 varied from 0.64 to 0.88 N·m, while the torque on No. A2 increased from 1.17 to 1.74 N·m, and the torque on No. A3 ranged from 1.06 to 1.71 N·m. In principle, torque is one of the energy input parameters in the drilling process and the rate of temperature increase at the front of the drilling tool was affected by torque variation.

### 4.2. Simulation Parameters and Results

#### 4.2.1. Simulation Parameters

The parameter settings used in the drilling simulations are listed in Table 5. It is worth mentioning that the size of the drill tool in this paper was determined by combining theory with experiment. The drill tool's size range was determined by theoretical calculation. Specifically, through theoretical calculation, the diameter of the drill tool range was determined, as was whether or not the performance of each size of the drill tool achieved its expected effect in the test. When the diameter of the drill is too small, the drill is more likely to easily break in the process of drilling, whereas when the diameter of the drill is too large, drilling speed is affected. The final size of the drill was determined according to the overall performance of the drill.

**Table 5.** Parameter.

| Parameters of the Particle System | Parameter |
| --- | --- |
| Effective thermal conductivity of particles (W/m·K) | 50.00 (A1), 50.00 (A1), 66.18 (A1) |
| specific heat capacity of a particle (J/kg·°C) | 228.95 (A1), 228.95 (A2), 406.70 (A3) |
| Particle radius (mm) | 0.8 mm, 1.6 mm, and 2.4 mm |
| Particle density (kg/m$^3$) | $3 \times 10^3$ |
| Shear modulus of particles (Pa) | $4 \times 10^7$ |
| Poisson's ratio of particles | 0.25 |
| Thermal conductivity of geometry (W/m·K) | 44.19 |
| Specific heat of geometry (J/kg·°C) | 544.00 |
| Geometric density (kg/m$^3$) | $7.85 \times 10^3$ |
| Shear modulus of geometry (Pa) | $8 \times 10^{10}$ |
| Poisson's ratio of geometry | 0.25 |
| Particle-particle friction coefficient | 0.50 |
| Particle-geometry friction coefficient | 0.48 |
| Coefficient of restitution | 0.50 |
| Inner diameter of drill pipe (mm) | 12.00 |
| External diameter of drill pipe (mm) | 17.00 |
| Thickness of drill pipe (mm) | 2.50 |
| Cross-sectional area of drill pipe (mm$^2$) | 113.83 |
| Rotational speed of drill (rpm) | 120.00 (A1), 250.00 (A2), 250.00 (A3) |
| Feed rate of the drill (m/s) | $1.03 \times 10^{-5}$ (A1), $2.10 \times 10^{-4}$ (A2), $9.50 \times 10^{-5}$ (A3) |
| Initial temperature (°C) | −139.60 (A1), −149.70 (A2), −97.20 (A3) |

### 4.2.2. Simulation Results

The difference in temperature fitting curves for each drill bit, comparing the experimental and simulation results, are plotted in Figures 9–11.

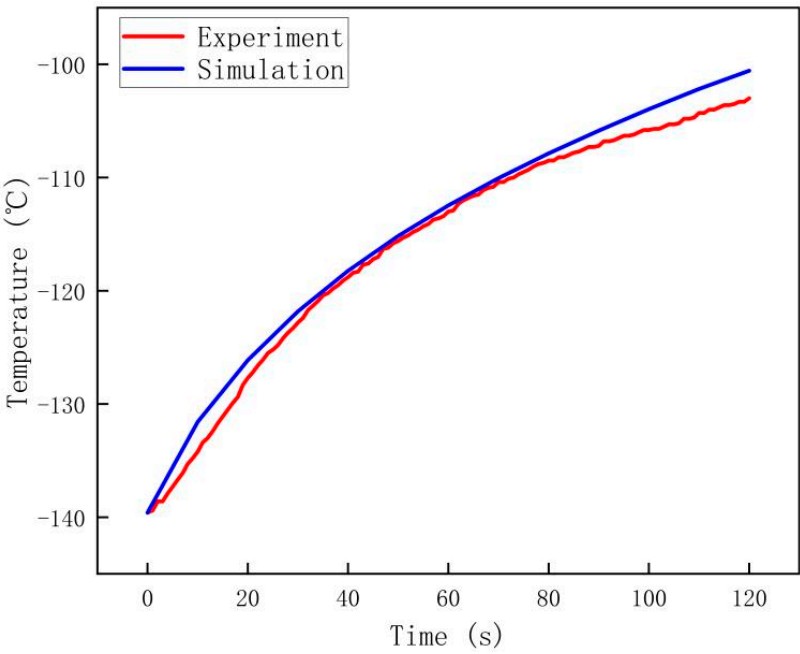

**Figure 9.** Drill bit temperature curve for No. A1 in experiment and simulation.

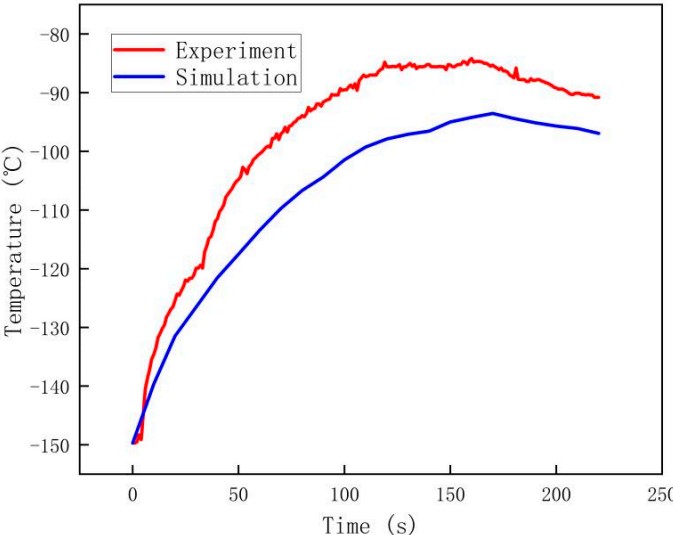

**Figure 10.** Drill bit temperature curve for No. A2 in experiment and simulation.

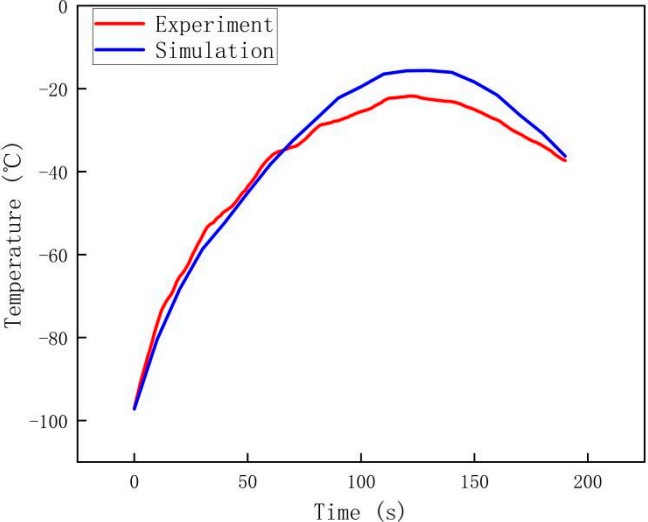

**Figure 11.** Drill bit temperature curve for No. A3 in experiment and simulation.

The curve-fitting method followed herein is described as follows: the temperature curves of the simulation and experimental results were regarded as points in n-dimensional space, wherein a smaller distance between the simulation and experimental values indicates the correspondence between the simulation and experimental results, as expressed in Equation (2):

$$R = 1 - \sqrt{\frac{\sum(y - y')^2}{\sum y^2}} \tag{2}$$

where R denotes the fitting degree of the curve; y denotes the experimental value; y′ denotes the simulation value.

As expressed in Equation (2), an R value proximate to 1 represents a higher fitting degree between the simulation and experimental results, and an R value approaching 0 indicates a lower fitting degree between the simulation and experimental results.

The temperature variations observed at the measurement points in the drilling experiment and simulation are comparatively presented in Table 6. Combined with the results in Figures 9–11, the maximum deviation between the temperature measurements from the experiment and simulation of Nos. A1, A2, and A3 was 120, 60, and 150 s and the temperature variation was 2.63, 13.32, and 5.43 °C, respectively. However, as observed

from the data in Table 4, the fitting degree of the three groups of simulations was above 0.9, demonstrating the reasonable accuracy of the simulation calculation model.

**Table 6.** Comparison of temperature variations at simulation and experimental measurement points.

| No. | A1 | A2 | A3 |
|---|---|---|---|
| Temperature increase in experiments (°C) | 36.60 | 58.80 | 59.11 |
| Temperature increase in simulation (°C) | 39.04 | 52.77 | 60.94 |
| Relative error of temperature rise | 6.67% | 10.26% | 3.10% |
| Maximum error of experimental and simulation (°C) | 2.63 | 13.32 | 5.43 |
| Curve-fitting degree | 0.99 | 0.90 | 0.92 |

*4.3. Analysis and Discussion*

In this study, the energy input of the drill power acts as the heat source for the drill and particles. The energy input distributes the generated thermal to the drill and the simulated lunar soil. During drilling, the heat distribution ratio between the drill tool and the simulated lunar soil varies with the speed of the drill tool, wear degree of the blade, and other factors. The preliminary allocation ratio was determined according to mechanical studies combined with early simulation experience; aiming at the change of particle heat distribution ratio between drill tool and simulated lunar soil. Based on the experimental data of No. A1, five sets of simulations were performed over a large range to compare the temperature increase in the drill bit with those recorded in the experiment.

The simulation and experimental data of the drill bit temperature in No. A1 under various distribution ratios are comparatively presented in Figure 12. Assuming that the heat distribution ratios between the drilling tool and simulated lunar soil particles were 5:5, 6:4, 7:3, 8:2, and 9:1, the final temperature rise error figures for the simulation and experimental temperature measurement points were 20.84%, 6.67%, 6.03%, 18.79%, and 29.78%, respectively. At the temperature measurement point, when the heat distribution ratio between the drill tool and simulated lunar soil was 7:3, the temperature rise error between the simulation and experiment was at its lowest. When the heat distribution ratio between the drilling tool and simulated lunar soil was 9:1, the temperature rise error between the simulation and experiment was the largest, and through the simulation under five sets of different distribution ratios, it can be seen that although the temperature of the drill bit changed after the end of drilling, the temperature rise error of the five sets of simulations was within 30% compared with that of the experiment at the drill bit. When there is an error between the thermal allocated by the drill or simulated lunar soil and the given allocation ratio, the effect is reduced by heat transfer. It can be observed that the calculation model has a certain degree of adaptability, and the calculation results are not invalid due to small fluctuations in the distribution ratio.

In NO. A2, the final temperature rise errors of the simulated temperature point and the experimental temperature point were 34.70%, 15.60%, 3.09%, 0.13% and 11.30%, respectively. In NO. A3, the final temperature rise errors of the simulated and experimental temperature points were 37.46%, 24.43%, 10.26%, 21.37% and 39.24%, respectively. It is worth mentioning that in NO. A2, although the error of the distribution ratio between drill tool and simulated lunar soil at 8:2 was smaller than that at 7:3, when the distribution ratio between drill tool and simulated lunar soil was 7:3, the trend of temperature change at the temperature measuring point in experiment and simulation was closer. The error of NO. A2 and NO. A3 is indeed larger than that of NO. A1. In subsequent studies, more in-depth research will be conducted on the heat distribution ratio of drilling tools and simulated lunar soil.

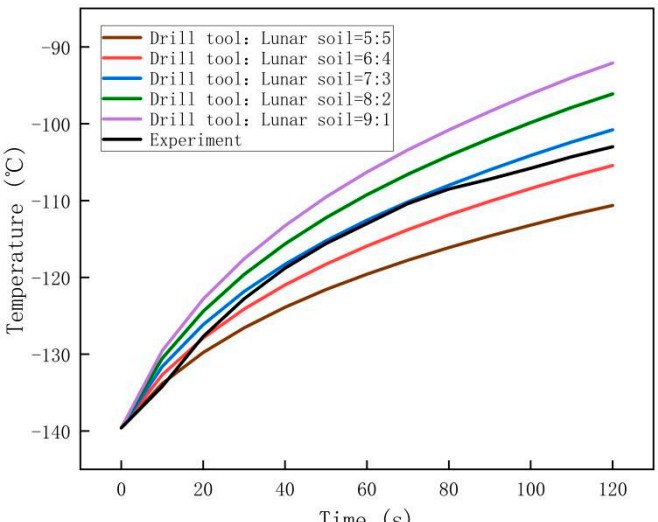

**Figure 12.** Comparison of simulation and experimental data regarding drill bit temperature in No. A1 under various distribution ratios.

The temperature distribution of the drill tool and the simulated lunar soil particles in No. A2 at various time intervals are portrayed in Figure 13, wherein the highest drill bit temperature was observed. The high-temperature particles of the simulated lunar soil were predominantly concentrated at the front of the drill bit and surrounding the drill pipe. In particular, the highest temperature of the particles was observed at the front of the drill bit. The high-temperature simulated lunar soil particles move upward continuously with the spiral wing of the drill tool, and eventually, the particles were discharged up until they reached the surface of the simulated lunar-soil-particle system.

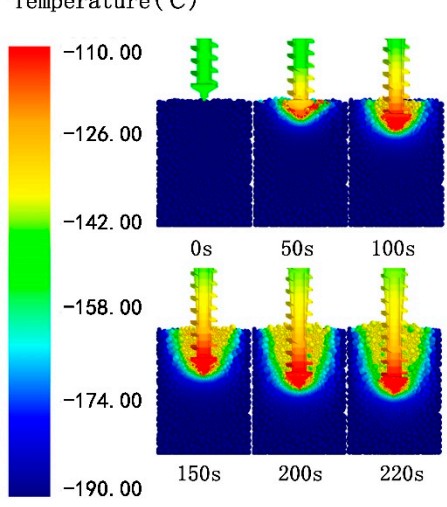

**Figure 13.** Temperature distribution of drill tool and simulated lunar soil particles in No. A2 at various intervals.

The velocity field of the particles in No. A2 along the Z-axis is presented in Figure 14, wherein the majority of the simulated lunar soil particles were slightly affected by the drilling process. The velocity of the simulated lunar soil particles near the drilling tool fluctuated significantly, and most particles traversed along the positive direction of the Z-axis toward the surface of the particle system. In the simulation, the high-velocity lunar soil particles were primarily concentrated near the spiral wing of the drilling tool. In particular, a small portion of the simulated lunar soil particles was influenced by the drilling direction, particle movement, and drill bit configuration. These particles traversed along the negative

direction of the Z-axis and were mainly concentrated near the drill bit. However, the number of these particles gradually declined with the increase in drilling depth, which is one of the reasons for the high temperature of the simulated lunar soil particles at the front of the drilling tool.

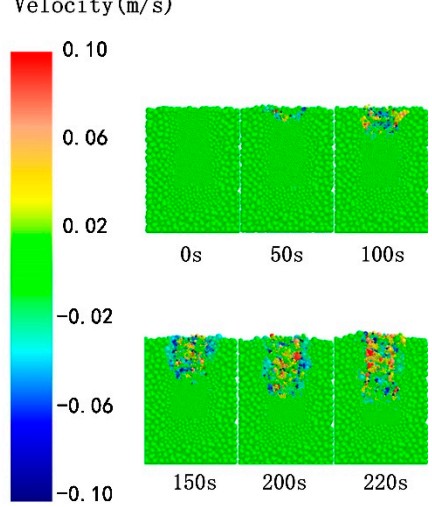

**Figure 14.** Velocity field of particles in No. A2 along Z-axis.

As observed in Figures 13 and 14, the simulated lunar soil particles surrounding the drilling tool were gradually elevated to the surface of the particle system along the spiral wing of the drilling tool, which makes it easier for the drilling tool to contact the particles with a lower temperature and effectively dissipate the thermal of the drilling tool; therefore, reducing the rate of its temperature increase. This can prevent a decline in drilling tool performance due to excessive temperature and preserve the characteristics of the acquired lunar soil samples.

## 5. Conclusions

1.  The error between the results of the discrete element simulation and experiments in terms of temperature increase was approximately 10%, indicating that the developed model can calculate the increase in drilling tool temperature with a certain applicability in the drilling process.
2.  Under the drilling conditions and reasonable considerations of this study, the maximum increase in the drill bit temperature was approximately 60 °C.
3.  The heat distribution ratio between the drill tool and the simulated lunar soil will change during drilling. The results indicated that the current calculation model exhibited a high adaptability, and the calculation results were not invalid due to fluctuations in the distribution ratio.
4.  In the simulations, the majority of the lunar soil particles near the drill tool traversed along the positive direction of the Z-axis, and the flow of the simulated lunar soil particles could effectively reduce the rate of temperature increase for the drill bit. Only a few particles near the drill bit traversed along the negative Z-axis, which is one of the reasons for the high temperature of the simulated lunar soil particles at the front of the drill tool.

**Author Contributions:** Conceptualization, J.C. (Jinsheng Cui); methodology, J.C. (Jinsheng Cui) and L.K.; software, L.K.; validation, W.Z., D.Z. and J.C. (Jiaqing Chang); writing—original draft preparation, L.K.; writing—review and editing, J.C. (Jinsheng Cui), W.Z., D.Z. and J.C. (Jiaqing Chang); funding acquisition, J.C. (Jinsheng Cui) and W.Z. All authors have read and agreed to the published version of the manuscript.

**Funding:** This research was funded by National Natural Science Foundation of China, grant number 51705092; Science and Technology Program of Guangzhou, grant number 202102020320 and 202102010461; 2021 Guangzhou University full-time graduate "Basic Innovation" project, grant number 2021GDJC-M28.

**Data Availability Statement:** The data presented in this study are available on request from the corresponding author.

**Conflicts of Interest:** The authors declare no conflict of interest.

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
