# Peer review of "Simulation of Drilling Temperature Rise in Frozen Soil of Lunar Polar Region Based on Discrete Element Theory"

_aerospace, doi:10.3390/aerospace10040368_

Round 1

Reviewer 1 Report

The authors attempt to explain the temperature and heat transfer between the drilling tool and simulated lunar soil simulant during the sampling. The application of DEM for this study was justified and experimentally compared.

Here are a few comments to address:

Page No: 5 – It is advised to explain the material (Lunar soil or Simulant) used for the experimental tests. If the lunar soil simulant was used, it is necessary to mention the type and give the properties of the lunar soil simulant. Clarify this.

The sample preparation in the test tank needs to be explained clearly. It is essential to mention at what relative density (loose or dense state) the sample was prepared and frozen. The denseness/compactness/packing of particles will influence the heat transfer.

Page No: 6 – It is advised to justify the reason behind selecting 5% and 10% water content and include references if anything is available.

Page No: 8 – It is also advised to clarify the selection of inner and outer dia of the drill. Want to know the effect on heat transfer by changing the dia of the drill.

Page No: 11 – The author has explained the simulation and experimental data of drill bit temperature in No. A1 for various distribution ratios. It is advised to give and explain the same for the remaining Nos. i.e. A2 and A3.

Author Response

Comment 1:

      Page No: 5 – It is advised to explain the material (Lunar soil or Simulant) used for the experimental tests. If the lunar soil simulant was used, it is necessary to mention the type and give the properties of the lunar soil simulant. Clarify this.

Response to comment 1:

      The materials and related properties of simulated lunar soil proposed by the reviewer and the preparation of simulated lunar soil samples are essential for the integrity of the article. We will add this content in the revision of the manuscript. Table 1 shows the minerals, particle size range and content of simulated lunar soil samples.

Table 1 The minerals, particle size range and content of simulated lunar soil samples

Mineral class

@Proportion of

Particle size range

Proportion of

Anorthosite(A)

@(70%)

0.025-0.05mm

31.568%

0.05-0.075mm

6.797%

0.075-0.1mm

10.545%

0.25-0.5mm

10.545%

0.5-1mm

10.545%

Basalt(B)

@(30%)

0.025-0.05mm

13.502%

0.05-0.075mm

2.920%

0.075-0.1mm

4.526%

0.25-0.5mm

4.526%

0.5-1mm

4.526%

      Lunar soil frozen soil samples shall be configured according to the following process:

  1. Weigh all kinds of particle size anorthoclase and basalt, into the oven for drying (more than 8 hours);
  2. According to the different material different particle size ratio configuration, put into the blender for uniform mixing;
  3. After the mixing of dry soil, the mixing water of samples should be allocated according to dry soil and different water content.
  4. After the completion of mixed water configuration, homogenized seal stand for 6 to 8 hours;
  5. Use a press to compact the sample five times to the required compactness;
  6. Sample the samples after compaction to verify the actual moisture content of the samples after preparation;
  7. Transfer the sample to the secondary refrigeration freezer (-80℃) for storage after 6-8 hours of primary refrigeration (-30℃), and the sample needs to undergo secondary refrigeration for 6-8 hours before use.

      Table 2 shows the effective thermal conductivity, specific heat capacity and compactness of the simulated lunar soil frozen soil particle system measured in the experiment.

Table 2 Experimental parameters of simulated lunar soil frozen soil Under normal pressure

Type

Parameter

Particle size range

0-1 mm

Sample temperature

93 K

Basic mineral

Pure dry soil sample

and mixed water sample

Moisture content

5wt%

10wt%

Density (g/cm3)

1.9

1.75

Measurement result

Effective thermal conductivity (W/(m·K))

0.8611

1.1397

Specific heat capacity

(J/(kg·â„ƒ))

228.95

270.6

Comment 2:

      Page No: 6 – It is advised to justify the reason behind selecting 5% and 10% water content and include references if anything is available.

Response to comment 2:

      The question about the selection of moisture content of frozen soil in polar region raised by the reviewer is based on literature search. Water content of frozen soil in the lunar polar region: According to ESA survey data, the water content of frozen soil in the lunar polar region is less than 11.9%[1]. According to our preliminary research and analysis results, the water content of lunar frozen soil in the permanently Shadowed area formed by meteorite impact craters at the lunar South Pole is between 5%-15% [2]. Therefore, after comprehensive consideration, simulated lunar soil with water content of 5% and 10% was studied in the experiment. Meanwhile, we have included a statement on the selection of frozen soil moisture content in the lunar polar region in Section 2.2.1 of the manuscript.

Reference:

[1] Magnani, P; Savoia,M; Hazan, D. Lunar icy soil sampling. The 64th International Astronautical Congress. Beijing, International Astronautical Federation, 2015, A3—2B—2.

[2] Liu, D.Y.; Wang, L.S.; Sun, Q.C.; Lai, X.M.; Liu, J.W. Experimental study on simulated lunar soil drilling of frozen soil in lunar polar region(in Chinese). J. Science Technology and Engineering, 2018, 18, 256—261.

Comment 3:

      Page No: 8 – It is also advised to clarify the selection of inner and outer dia of the drill. Want to know the effect on heat transfer by changing the dia of the drill.

Response to comment 3:

      In the experiment, we considered flat edge, cone edge, curved edge and so on for the drill bit configuration. For flat edge, chip fluidity is not good in drilling test, which is not conducive to sampling. Curved edge drilling effect is good, but the processing difficulty is higher. Conical edge has the best comprehensive performance in the experiment. In our team we have professionals who do research on drill tools. Through the theoretical method to determine the scope of drilling tool size, and then through the experiment to determine the final size. In the experiment, it is possible to fracture if the diameter of the drill is too small, and the speed of drilling will be affected if the diameter of the drill is too large. The inside diameter of the drill pipe in this paper is 12mm and the outside diameter is 17mm. Table 5 in the manuscript, I mistakenly wrote the outside diameter and inside diameter of the drill tool as the radius. I'm sorry if I have influenced your judgment. Small lifting Angle of drilling tool is not conducive to chip removal, and large lifting Angle will slide the sample, which is not conducive to sampling. No special experiments have been conducted on the effect of different diameter bits on the drill. Of course, your suggestion has brought us a new research idea, and we will further study this aspect in the subsequent experiments. And the selection of drill size is described in Section 4.2.1.

Comment 4:

      Page No: 11 – The author has explained the simulation and experimental data of drill bit temperature in No. A1 for various distribution ratios. It is advised to give and explain the same for the remaining Nos. i.e. A2 and A3.

Response to comment 4:

      The reviewer proposed that the two groups of NO.A2 and A3 simulated the temperature changes of the drill under different distribution ratios, and made the following explanation. In the simulation, the drill and simulated lunar soil distribution ratio is used to calculate the heat generated at each time step through a proportional distribution between the drill and simulated lunar soil particles in contact with the drill at this time step. In our group, there are researchers who do this kind of research, and in this paper the allocation ratio is the data that we got from the researchers who did this study and used it in our calculations. In fact, the change in the drill tool and the simulated lunar soil from 7:3 to 6:4 is a very big change. We can see from the manuscript that when the heat distribution ratio is 5:5 and 9:1, the simulation results have already deviated greatly from the experimental results. The purpose of setting this distribution ratio simulation is to prove that the distribution ratio does fluctuate in the simulation process, but the range of fluctuations is very small. In the simulation process, the heat distribution between the drill tool and the simulated lunar soil calculated by a constant has little influence on the results. In the simulation, the mutual heat transfer between the drill tool and the simulated lunar soil will balance out part of the error. But there are limits to how much balance can be achieved.

      For the A2 and A3 groups, we simulated different allocation ratios between 5:5 and 9:1. Simulation results are shown in Figure 1 and Figure 2 below. Based on the simulation and experimental data, it is assumed that the heat distribution ratio between the drill tool and the simulated lunar soil particles is 5:5, 6:4, 7:3, 8:2 and 9:1, respectively. In NO.A2, the final temperature rise errors of the simulated temperature point and the experimental temperature point are 34.70%, 15.60%, 3.09%, 0.13% and 11.30%, respectively. In NO.A3, the final temperature rise errors of the simulated and experimental temperature points were 37.46%, 24.43%, 10.26%, 21.37% and 39.24%, respectively.

      It is worth mentioning that in NO.A2, although the error of the distribution ratio between drill tool and simulated lunar soil at 8:2 was smaller than that at 7:3, when the distribution ratio between drill tool and simulated lunar soil was 7:3, the trend of temperature change at the temperature measuring point in experiment and simulation was closer. The error of NO.A2 and NO.A3 is indeed larger than that of NO.A1. In subsequent studies, more in-depth research will be conducted on the heat distribution ratio of drilling tools and simulated lunar soil. In Section 4.3 of the paper, errors of simulation and experiment at temperature measurement points of NO.A2 and NO.A3 under different distribution ratios are added.

 Figure 1. Comparison of simulation and experimental data of drill bit temperature in No. A2 under various distribution ratios

Figure 2. Comparison of simulation and experimental data of drill bit temperature in No. A3 under various distribution ratios

Reviewer 2 Report

This is a highly interesting paper describing well-conceived and well-executed research on an important topic. I recommend it for publication with only two items I believe the authors should discuss briefly.

1) Equation 1 for heat conductance in the icy soil is linear in temperature. If the ice completely fills the spaces between the grains then this is probably correct.

However, measurements of granular materials that have significant porosity (the spaces between grains not filled by ice) show a temperature dependence that is a linear term plus a quadratic term. The linear term is for solid conduction while the quadratic term is for radiative transfer through the pore spaces between the grains. 

The paper you cited for this approach says that radiation can be ignored because the temperature was low in their work (25 degC) and they used only small temperature differences. In your work, the temperatures are even lower but the temperature variation is quite large (60 deg C). With T^4 dependence, a 60 degree change can be very significant. I did a quick calculation using thermal conductances measured on Apollo lunar soils and this suggests there will be significant change in the effective conductance coefficient over this temperature range. 

I recognize that your simulations matched the experiments very well, so I am guessing the approximation was adequate, but I do not know why. Please say something about this to explain why your approximation is adequate since I think other readers may wonder, as well.

2) Also, I think you should report how the icy regolith was prepared. Was water added in liquid form and then the mixture was frozen together, or was ice crushed and then mixed into the soil in solid granular form, or some other process?  For other researchers to be able to replicate your results I think the method should be reported.

Author Response

Comment 1:

      Equation 1 for heat conductance in the icy soil is linear in temperature. If the ice completely fills the spaces between the grains then this is probably correct.

      However, measurements of granular materials that have significant porosity (the spaces between grains not filled by ice) show a temperature dependence that is a linear term plus a quadratic term. The linear term is for solid conduction while the quadratic term is for radiative transfer through the pore spaces between the grains.

      The paper you cited for this approach says that radiation can be ignored because the temperature was low in their work (25 degC) and they used only small temperature differences. In your work, the temperatures are even lower but the temperature variation is quite large (60 degC). With T^4 dependence, a 60 degree change can be very significant. I did a quick calculation using thermal conductances measured on Apollo lunar soils and this suggests there will be significant change in the effective conductance coefficient over this temperature range.

      I recognize that your simulations matched the experiments very well, so I am guessing the approximation was adequate, but I do not know why. Please say something about this to explain why your approximation is adequate since I think other readers may wonder, as well.

Response to comment 1:

      First of all, it needs to be explained that the relative compaction of simulated lunar soil required by the experiment in the manuscript is 99%.

      Secondly, regarding the effects of simulated lunar soil radiation, the heat transfer between particles calculated according to Eq. (1) only mainly represents the heat conduction between particles, and it seems that convection and radiation are not considered at all. On the one hand, the main reason is that the heat transfer between particles plays a leading role. On the other hand, in order to simplify the model, all heat transfer processes between particles are generally simplified as heat conduction in DEM, which is called effective heat conduction and characterized by effective thermal conductivity (ETC). In the subsequent calibration process of particle parameters, the goal is to make the simulated particle set similar to the measured effective thermal conductivity of the actual simulated lunar soil. In fact, for a particle system, the experimental thermal conductivity is the ETC under the experimental conditions. To sum up, considering the simplification and error of the model, the present study focused on the ETC in the investigation of heat transfer through a granular assembly.

      Finally, we only use the good fitting degree of drill bit temperature in experiment and simulation to judge the good fitting degree of the whole particle system and the whole drill tool. There is a certain error. There is an error in establishing particle system only by heat transfer equivalence. In the subsequent study, we will conduct experimental tests to add radiation to the particle system and observe its effects.

Comment 2:

      Also, I think you should report how the icy regolith was prepared. Was water added in liquid form and then the mixture was frozen together, or was ice crushed and then mixed into the soil in solid granular form, or some other process? For other researchers to be able to replicate your results I think the method should be reported.

Response to comment 2:

      The reviewer raised questions related to the preparation of simulated lunar soil, which was also mentioned by another reviewer. I will add this part of information in the process of revising the manuscript later. This is essential for the integrity of the article. Table 1 shows the minerals, particle size range and content of simulated lunar soil samples.

Table 1 The minerals, particle size range and content of simulated lunar soil samples

Mineral class

@Proportion of

Particle size range

Proportion of

Anorthosite(A)

@(70%)

0.025-0.05mm

31.568%

0.05-0.075mm

6.797%

0.075-0.1mm

10.545%

0.25-0.5mm

10.545%

0.5-1mm

10.545%

Basalt(B)

@(30%)

0.025-0.05mm

13.502%

0.05-0.075mm

2.920%

0.075-0.1mm

4.526%

0.25-0.5mm

4.526%

0.5-1mm

4.526%

      Lunar soil frozen soil samples shall be configured according to the following process:

  1. Weigh all kinds of particle size anorthosite and basalt, into the oven for drying (more than 8 hours);
  2. According to the different material different particle size ratio configuration, put into the blender for uniform mixing;
  3. After the mixing of dry soil, the mixing water of samples should be allocated according to dry soil and different water content.
  4. After the completion of mixed water configuration, homogenized seal stand for 6 to 8 hours;
  5. Use a press to compact the sample five times to the required compactness;
  6. Sample the samples after compaction to verify the actual moisture content of the samples after preparation;
  7. Transfer the sample to the secondary refrigeration freezer (-80℃) for storage after 6-8 hours of primary refrigeration (-30℃), and the sample needs to undergo secondary refrigeration for 6-8 hours before use.

      Table 2 shows the effective thermal conductivity, specific heat capacity and compactness of the simulated lunar soil frozen soil particle system measured in the experiment.

Table 2 Experimental parameters of simulated lunar soil frozen soil

Type

Parameter

Particle size range

0-1 mm

Sample temperature

93 K

Basic mineral

Pure dry soil sample

and mixed water sample

Moisture content

5wt%

10wt%

Density(g/cm3)

1.9

1.75

Measurement result

Effective thermal conductivity(W/(m·K))

0.8611

1.1397

Specific heat capacity

(J/(kg·â„ƒ))

228.95

270.6

Round 2

Reviewer 1 Report

Dear Authors,

                      I am satisfied with your response to my comments and recommend publication.